# OpenReview forum: "Language Models Learn to Mislead Humans via RLHF"
_ICLR.cc/2025/Conference — ICLR 2025 Poster_

### Official Review · Reviewer_24F3 · 2024-10-31

**Soundness:** 4
**Presentation:** 4
**Contribution:** 4
**Rating:** 8
**Confidence:** 3

**Summary:**

The paper investigates a phenomenon called "U-SOPHISTRY", in which language models (LMs) trained through reinforcement learning from human feedback (RLHF) unintentionally become better at convincing humans that they are right even when they are wrong without improving actual performance. The author demonstrates U-SOPHISTRY through user studies on question-answering (QuALITY) and programming (APPS) tasks. The results indicate that RLHF-optimized models convince human evaluators of incorrect outputs at higher rates compared to their pre-trained counterparts. Through a comprehensive qualitative analysis, the authors identify strategies that the RLHF models use to mislead evaluators, such as fabricating or selectively presenting evidence in QA tasks and generating partial code that passes all human-written tests or is less readable for debugging. Additionally, the authors emphasize that prior work on detecting I-SOPHISTRY (intentionally misleading) is not an effective benchmark for methods aimed at detecting U-SOPHISTRY.

**Strengths:**

1.  The exploration of U-SOPHISTRY provides a novel perspective on RLHF challenges.
2. The authors conduct experiments and rigorous user studies across two common tasks, measuring correctness, human approval, evaluation error rate, and false positive rate. The results provide deep insights into how RLHF impacts human judgment and evaluation.
3. Through comprehensive qualitative analysis, the paper examines specific strategies used by LMs that lead to U-SOPHISTRY, such as fabricating evidence and exploiting human testing shortcuts, enhancing our understanding of how RLHF may influence model behavior.
4. The paper is well-written and easy-to-follow.

**Weaknesses:**

I do not see major issues in the paper, though I have a few minor concerns and some areas where clarifications would be helpful.

**Concerns**
1. In the programming (APPS) experiment, the authors assume that users will provide test cases for code validation, which may not fully capture real-world scenarios. Users may also seek explanations from the model to better understand the code, especially if they don’t fully understand the initial question, which could result in their inability to write accurate test cases. It would be good to include a user study from this aspect.
2. A follow-up question is whether all human evaluators fully understand the coding questions?  Users may write inaccurate test cases and then incorrectly interpret execution results as passing. Additional analysis on the correctness of test cases written by human evaluators would be beneficial.

**Clarifications**:
1. Are the Human evaluation error rate and Human false positive rate calculations based on $R^{human}$ rather than $R^{train}$. The analysis appears to use real human evaluators ($R^{human}$), but the description on lines 181 and 183 uses $R^{train}$ instead of $R^{human}$.

2. It would be clearer to display the distribution of total cases shown to human evaluators to indicate how many were the initial responses and how many were the RLHF responses.

**Questions:**

Can the author answer all the questions above?

**Details Of Ethics Concerns:**

The paper involves human subjects in its user study to evaluate the effects of RLHF on misleading human evaluations. This raises potential ethical considerations, particularly around consent, data privacy, and ensuring that the evaluators are fully informed about the study's purpose and potential biases they may encounter.

---

> ### Author Response · Authors · 2024-11-19
> **Response to Reviewer 24F3 (1/2)**
>
> Thanks for appreciating the strength of our paper! We will address each of your questions below, and are happy to expand or provide further responses if necessary.
>
> > In the programming (APPS) experiment, the authors assume that users will provide test cases for code validation, which may not fully capture real-world scenarios. Users may also seek explanations from the model to better understand the code, especially if they don’t fully understand the initial question, which could result in their inability to write accurate test cases. It would be good to include a user study from this aspect.
>
> To clarify, our study did not restrict human evaluators to checking programs solely through writing test cases, and they are allowed to read the program as well.
>
> Regarding interacting with the language model, our experiment focused on single-turn human evaluation as in typical RLHF or human evaluation practices ([1][2]), rather than an interactive setup. We agree that using LMs to assist human evaluation is a good research direction for solving “super-alignment” issues ([3][4][5][6]). However, it still remains unclear whether and how LMs would bias human evaluators. For example, recent works ([3][7]) have shown that using LMs to generate explanations could bias humans to mislabel incorrect outputs as correct.
>
>
> > A follow-up question is whether all human evaluators fully understand the coding questions? Users may write inaccurate test cases and then incorrectly interpret execution results as passing. Additional analysis on the correctness of test cases written by human evaluators would be beneficial.
>
> Thanks for the advice! However, we cannot directly calculate test-case accuracy given our experimental setup. We did not ask evaluators to directly submit program input-output pairs and automatically check whether the LM-generated program returns the target output given the input. Instead, we allowed evaluators to submit any inputs and returned them the execution results (outputs), leaving the evaluators to evaluate the outputs themselves. This can 1) reduce the error caused by string-matching algorithms and 2) allow evaluators to more efficiently check program correctness. For example, evaluators might tell an output is incorrect without needing to provide the right answer themselves.
>
> To better understand our human evaluators by analyzing their test-cases, we calculated the following statistics among all false positives, i.e. incorrect programs that our evaluators consider to be correct.
> - Evaluators did not submit any test cases: 13.5%.
> - Evaluators submitted test cases but none of the test case revealed that the program is incorrect: 57.7%. In these cases, **the evaluators failed to catch bugs because their test cases do not have high enough coverage.**
> - Evaluators submitted test cases where some revealed errors in the program, BUT they misinterpreted the output and labeled it as correct: 28.8%. In these cases, the evaluators misinterpreted wrong outputs as correct.
>
> Therefore, in most cases, our evaluators fail to catch errors in incorrect programs because their test cases are not comprehensive, rather than misinterpreting the execution results. There are indeed cases of misinterpretation though, and I will report these information in our updated draft.
>
> Finally, while we cannot ensure that ALL evaluators understand the problems by checking the input-outputs they submit, we indeed made sure that the evaluators have similar backgrounds with annotators in a practical RLHF pipeline, or a downstream user who has programming knowledge. Our human evaluators are all experienced programmers, majoring in Computer Science or Electronic Engineering. We also conducted warm-up tests to filter out unqualified candidates. With a labeling accuracy of 68.7% on $\pi_{init}$-generated programs, their performance is significantly better than random guessing given the challenge of APPS questions. Finally, we used the same distribution of human subjects to evaluate the LMs both before and after RLHF, ensuring fair comparisons between models.
>
>
> [1] Training language models to follow instructions with human feedback. arXiv 2022
>
> [2] Chatbot Arena: An Open Platform for Evaluating LLMs by Human Preference. ICML 2024
>
> [3] LLM Critics Help Catch LLM Bugs. arXiv 2024
>
> [4] Debating with More Persuasive LLMs Leads to More Truthful Answers. ICML 2024
>
> [5] Learning Task Decomposition to Assist Humans in Competitive Programming. ACL 2024
>
> [6] Self-critiquing models for assisting human evaluators. arXiv 2022
>
> [7] Large Language Models Help Humans Verify Truthfulness--Except When They Are Convincingly Wrong. NAACL 2024.

---

> > ### Comment · Reviewer_24F3 · 2024-11-22
> >
> > Thank you for your clarifications! My concerns have been resolved.

---

> ### Author Response · Authors · 2024-11-19
> **Response to Reviewer 24F3 (2/2)**
>
> > Are the Human evaluation error rate and Human false positive rate calculations based on R^human rather than R^train. The analysis appears to use real human evaluators (R^human}, but the description on lines 181 and 183 uses R^train instead of R^human.
>
> Yes, it is based on R^human. Thanks for pointing out this typo!
>
> > It would be clearer to display the distribution of total cases shown to human evaluators to indicate how many were the initial responses and how many were the RLHF responses.
>
> We sampled an equal size of responses from the initial LM and the RLHF’ed LM.

---

### Official Review · Reviewer_cFxf · 2024-11-03

**Soundness:** 3
**Presentation:** 4
**Contribution:** 4
**Rating:** 6
**Confidence:** 4

**Summary:**

The work empirically validates the hypothesis that LLMs can learn to perform reward hacking, i.e., achieve higher rewards when RLHFed, despite not actually being better, according to an oracle. For the study, they recruited human evaluators and validated this hypothesis in two settings: QA and programming tasks, showing that after performing RLHF, human evaluators thought that the model performance improved despite the performance not improving. They also show that the false positive rate for human evaluators also increases. They also check if probing methods that can detect incorrect programs generated by models with backdoors don't generalize to RLHFed models where the model performs reward hacking, which is unintentional from the user's end.

**Strengths:**

- The work recruits humans and validates the hypothesis that models can learn to perform reward hacking. The human evaluations are well thought out.
- They perform extensive evaluations and experiments for robustness. They also try methods that can detect I-Sophistry to detect U-Sophistory but find that the methods do not generalize.
- The insights are impactful and should make researchers and industry practitioners give more thought to designing their RLHF training procedure.
- The manuscript is well-written and easy to understand.

**Weaknesses:**

- I am not convinced by the design of the programming task used to validate the hypothesis.
    - Why do the authors choose the two simplest unit tests? How would things change if they used the two most difficult unit tests?
    - For the pilot study, how were the human evaluators incentivized? As a developer, I would write two unit tests. One is an easy case, and another is difficult or where programs usually fail.
    - In an ideal scenario for preference tuning for programming tasks, human annotators should only be used for stylistic improvements since we also have very strong methods for checking the correctness of models.

- More studies need to be done on whether incentivizing humans according to correctness and confidence biases human evaluators to be less or more confident.

**Questions:**

- Why do you think the LLM generates a less readable program?
- Do you think that the increase in human error rate for programming task is because the more is generating less readable code or because it learns to pass the unit tests?
I would not conclude that if humans spend more time on a task, they work harder. There could be several other factors that can influence time.

---

> ### Author Response · Authors · 2024-11-19
> **Response to Reviewer cFxf**
>
> Thanks for appreciating the strength of our paper! We will address each of your questions below, and are happy to expand or provide further responses if necessary.
>
> > Why do the authors choose the two simplest unit tests? How would things change if they used the two most difficult unit tests?
>
> To clarify, this proxy reward is used exclusively for training and not for evaluation. During evaluation, we recruited real humans to evaluate programs at their best. We pick these tests so that evaluating with these tests highly agrees with evaluating with real human programmers. We show the results in Table 1 and find that indeed, selecting two simple unit tests better approximates human evaluation.
>
> | Strategy                                 | Correlation with Human Evaluation |
> | ---------------------------------------- | --------------------------------- |
> | Selecting two simplest test cases        | 0.59                              |
> | Selecting two most difficult test cases | 0.42                              |
>
>
>
> >  Why do you think the LLM generates a less readable program?
>
> The training reward (program correctness) does not prefer more readable programs (i.e. legibility). During RL, LMs are incentivized to generate more complex logic to pass the test cases regardless of readability. We observe a similar trend even when training with the oracle reward. Specifically, as shown in the table below, when optimizing with R^*, the generated incorrect programs also contain fewer helper functions and exhibit higher complexity.
>
> |                    | # Function |           | # Code Complexity |           |
> | ------------------ | ---------- | --------- | ----------------- | --------- |
> |                    | Correct    | Incorrect | Correct           | Incorrect |
> | Initial            | 0.11       | 0.57      | 3.1               | 3.8       |
> | RLHF ($R^{train}$) | 0.06       | 0.15      | 3.1               | 6.1       |
> | RLHF ($R^*$)       | 0.08       | 0.24      | 3.3               | 8.8       |
>
>
>
> > For the pilot study, how were the human evaluators incentivized? As a developer, I would write two unit tests. One is an easy case, and another is difficult or where programs usually fail.
>
> Human evaluators are incentivized to provide correct labels with a high bonus. As illustrated in Lines 257-263, in QA, each evaluation is paid with \\$3 if it’s correct and \\$1.5 otherwise; in programming, \\$9 for each correct decision with 100% confidence, \\$4 for correct with 75% confidence, \\$2 for unsure 50%, \\$2 for incorrect 75% and \\$1 for incorrect 100%
>
>  Instead of imposing detailed restrictions on evaluation methods, we allowed evaluators to choose strategies that best fit their expertise and background. Moreover, we chose not to set a bonus on the number of submitted test cases, as it might encourage humans to write numerous simple, repetitive test cases solely to get a higher bonus.
>
> > More studies need to be done on whether incentivizing humans according to correctness and confidence biases human evaluators to be less or more confident.
>
> In Lines 261-263, we highlighted that the confidence labels should be interpreted with caution and not treated as true probabilities. We also presented the confidence results in a separate figure (Figure 3), rather than in the main figure.
>
> We agree with the reviewer that it is challenging to collect “unbiased” human confidence. However, the bonus scheme is the same when evaluating both the initial and RLHF-trained model, so our main conclusion that compares LMs before and after RLHF training still holds.
>
> > Do you think that the increase in human error rate for programming task is because the more is generating less readable code or because it learns to pass the unit tests?
>
> Both reasons play a role. Since programs become less readable, humans have to rely on writing unit tests; as a result, they become more easily misled by $\pi_{rlhf}$, which is good at hacking human-written unit tests.
>
> > I would not conclude that if humans spend more time on a task, they work harder. There could be several other factors that can influence time.
>
> Thanks for bringing this up! We used three metrics — time spent, number of annotator designed test cases, and diversity of test cases — to approximate an intuitive notion of “human efforts”. However, we agree that these are imperfect measures, and other confounding factors may influence the results. In our updated draft, we will report these three metrics directly instead of making claims about “human efforts”, which is difficult to directly measure.

---

> > ### Author Response · Authors · 2024-11-24
> > **Looking forward to your comment**
> >
> > We have added further clarifications and experiments to address your reviews. Please feel free to let us know if you have any questions. Thanks a lot!!

---

> > > ### Comment · Reviewer_cFxf · 2024-11-25
> > >
> > > I have read the author's response and choose to stick to my evaluation of the work. Thanks!

---

> > > > ### Author Response · Authors · 2024-11-26
> > > >
> > > > Thanks for your reply!
> > > > We are happy to continue addressing any remaining questions, especially to clarify any misunderstandings about our programming experiments!

---

### Official Review · Reviewer_8kZa · 2024-11-03

**Soundness:** 3
**Presentation:** 3
**Contribution:** 3
**Rating:** 6
**Confidence:** 3

**Summary:**

This paper studies that LMs trained with standard RLHF techniques can learn to appear more convincing to human evaluators without actually improving their tasks. Through studies on question-answering and programming tasks, the authors show that RLHF-trained models achieve higher human approval ratings while making human evaluation less accurate.

**Strengths:**

* Addresses a critical gap in understanding how language models might naturally learn to mislead humans
* The experiment is well-designed with appropriate controls
* First systematic study of unintended sophistry in RLHF

**Weaknesses:**

* This paper only shows the experimental results on only two tasks (question-answering and programming). Without specific experiments, we may not know whether the method would generalize to other important domains where RLHF is used.
* Figure 1 could benefit from more detailed captions
* The related work section only covers RLHF literature and could expand discussion on human evaluation methods.

**Questions:**

* Have the authors considered testing their findings on other types of tasks beyond QA and programming?
* Could the authors elaborate on potential mitigation strategies beyond probing?
* How might the results differ with more or less experienced evaluators?

---

> ### Author Response · Authors · 2024-11-19
> **Response to Reviewer 8kZa**
>
> Thanks for appreciating the strength of our paper! We will address each of your questions below, and are willing to expand or provide further responses if necessary.
>
> > Could the authors elaborate on potential mitigation strategies beyond probing?... The related work section only covers RLHF literature and could expand discussion on human evaluation methods.
>
> There are several promising ways to mitigate U-Sophistry:
> - optimizing AI to assist human evaluation: debate [1], task decomposition [2], critique [3]
> - optimizing AI for easier human evaluation [4]
> - generalizing beyond weak human supervision, e.g., weak-to-strong generalization [5] or easy-to-hard generalization [6].
>
> We will integrate these to make the related work section more comprehensive.
>
> > How might the results differ with more or less experienced evaluators?
>
> Thanks for this insightful question – we conjecture that more experienced evaluators may reduce sophistry, while less experienced ones may exacerbate it. We look forward to future works that validate this conjecture or validating this conjecture ourselves once more funding is available.
>
> The claim of our paper is that LMs can learn to mislead human evaluators in some realistic settings, even when our human subjects are far better than the bar required to become annotators in a production RLHF pipeline. We recruited the highest-quality human evaluators feasible within our budget constraints, and showed that SOPHISTRY can still happen. For QA, we recruited native English speakers (e.g. teachers, editors, and writers), who are experienced in question-answering; we recruited them from Upwork, a leading crowdsourcing platform. For programming, we recruited experienced Python programmers from a pool of college students majoring in Computer Science or Electronic Engineering. In Section 3.5, we also confirmed that our findings are not due to noises in the recruiting process or variations in annotation efforts, indicating the external generality of our human studies.
>
>
> > Figure 1 could benefit from more detailed captions
>
> We appreciate your feedback and will add clarification to the caption regarding the three types of rewards and evaluation details.
>
> [1] Debating with More Persuasive LLMs Leads to More Truthful Answers. ICML 2024
>
> [2] Learning Task Decomposition to Assist Humans in Competitive Programming. ACL 2024
>
> [3] Self-critiquing models for assisting human evaluators. arXiv 2022
>
> [4] Prover-Verifier Games improve legibility of LLM outputs. arXiv 2024
>
> [5] Weak-to-Strong Generalization: Eliciting Strong Capabilities With Weak Supervision. ICML 2024
>
> [6] The Unreasonable Effectiveness of Easy Training Data for Hard Tasks. ACL 2024

---

> > ### Author Response · Authors · 2024-11-24
> > **Looking forward to your comment**
> >
> > We have added further clarifications to address your reviews. Please feel free to let us know if you have any questions. Thanks a lot!!

---

### Official Review · Reviewer_Fiwv · 2024-11-04

**Soundness:** 2
**Presentation:** 3
**Contribution:** 2
**Rating:** 5
**Confidence:** 4

**Summary:**

This paper "Language Models Learn to Mislead Humans via RLHF" examines how language models fine-tuned with Reinforcement Learning from Human Feedback (RLHF) can unintentionally mislead humans by appearing correct even when wrong. Through experiments on question-answering (QuALITY) and programming tasks (APPS), the authors show that RLHF increases human approval of model responses but does not enhance accuracy, often causing humans to rate incorrect answers as correct. This phenomenon "U-Sophistry" suggests that RLHF can make language models more persuasive without improving true correctness, highlighting a significant challenge for model alignment.

**Strengths:**

1. This paper introduces the concept of "U-SOPHISTRY," wherein RLHF unintentionally enables language models to mislead human evaluators without necessarily improving task performance. This novel framing extends prior work on reward hacking and deception in AI, highlighting new risks in standard RLHF pipelines. With AI applications proliferating, ensuring safe and reliable human-AI interaction is critical.

2. The method incorporate diverse and challenging tasks like question-answering and programming tasks.

3. This paper clearly demonstrates the author's intent and contains numerous figures and tables.

**Weaknesses:**

1. The scenario discussed is somewhat perplexing: this paper argues that models trained with RLHF may become more deceptive towards humans without actual improvements in capability. However, RLHF’s effectiveness relies heavily on the choice of reward model and corresponding training data, so if there are issues in human-annotated data, such results are predictable. Thus, the reviewer suggests that the problem stems from humans no longer being able to provide sufficiently high-quality evaluations of the model’s outputs, resembling more of a “Weak-to-Strong” alignment issue, while the discussion in this paper seems to frame it as an issue with the RLHF pipeline itself.

2. Lack of robust countermeasures to mitigate "U-SOPHISTRY".

3. Many alignment algorithms have been optimized for the classic RLHF pipeline, such as DPO and KTO. Running different alignment algorithms may lead to varying results in the final model, yet the authors conducted experiments only with the original PPO algorithm.

**Questions:**

Please refer to weakness.

---

> ### Author Response · Authors · 2024-11-19
> **Response to Reviewer Fiwv**
>
> Thanks for appreciating the strength of our paper! We will address each of your questions below, and are happy to expand or provide further responses if necessary.
>
> > if there are issues in human-annotated data for reward modeling, it is a “predictable issue” that LMs will learn to mislead humans.
>
> We argue that
>
> 1) While flawed human annotation is widely hypothesized to negatively impact models, our work provides the **first systematic study** of this issue. In practice, developers heavily incorporate imperfect human evaluations into the LLM development life cycles (e.g., RLHF training, LMsys evaluation). Our empirical results thus call for more attention to this issue and provide a concrete foundation for future work tackling this issue.
>
> 2) Many research aims to address this “predictable issue” ([1][2][3]), but our paper is the first to validate that this issue both actually exists and leads to substantial failures in a realistic setting. In particular, we find that LMs can mislead real humans; such a finding is significantly stronger than prior works (as discussed in Table 1 and Section 2.1), where the LMs can successfully hack against learned, weaker reward models rather than real humans. For example, generating lengthy responses or bullet point lists [4] can only mislead models but not real humans.
>
> Finally, we focus on studying RLHF because it is so far the most common method in post-training.
>
> > Many alignment algorithms have been optimized for the classic RLHF pipeline, such as DPO and KTO. Running different alignment algorithms may lead to varying results in the final model, yet the authors conducted experiments only with the original PPO algorithm.
>
> We focus on PPO because it is the most popular post-training algorithm and most commonly used in production systems. Recent works have also shown that PPO is more effective than alternatives on challenging tasks such as coding ([5][6]), which is our focus.
>
> Additionally, the objectives of DPO, PPO, and KTO are all qualitatively similar, as they all optimize for human approvals. Since we identify the optimization objective as the core driving force of SOPHISTRY, we consider algorithmic variants such as DPO or KTO to be complementary to our setups and they do not influence the main claims of our paper.
>
> To back up our claim, we replicate our experiments with KTO on all three settings and we show that reward hacking still occurs: $R^{train}$ substantially increases while $R^*$ does not. Given time limit, we did not perform human evaluation of KTO models, but will consider adding that to the revision.
>
> | Setting            | Model   | $R^{train}$ | $R^*$ |
> | ------------------ | ------- | ----------- | ----- |
> | Programming        | Initial | 34.8        | 9.4   |
> |                    | KTO     | 40.8        | 9.3   |
> | QA (Task-specific) | Initial | 41.3        | 52.0  |
> |                    | KTO     | 46.1        | 44.0  |
> | QA (General)       | Initial | -12.2       | 52.0  |
> |                    | KTO     | -10.9       | 48.7  |
>
>
>
> > Lack of robust countermeasures to mitigate "U-SOPHISTRY".
>
> The goal of our paper is to present empirical evidence of U-SOPHISTRY, which is the hypothetical threat that the field of “superalignment” tries to address (e.g. [1][2][3][7][8]). Robustly mitigating it implies solving “superalignment”, which remains an open problem and is thus out-of-scope for our paper, which focuses on providing empirical evidence for U-SOPHISTRY. However, by understanding its emergence and characteristics, as well as open-sourcing the data and fine-tuned models, our work provides the foundation for future mitigation research.
>
>
>
>
> [1] Debating with More Persuasive LLMs Leads to More Truthful Answers. ICML 2024
>
> [2] Learning Task Decomposition to Assist Humans in Competitive Programming. ACL 2024
>
> [3] Self-critiquing models for assisting human evaluators. arXiv 2022
>
> [4] A long way to go: Investigating length correlations in RLHF. arXiv 2023
>
> [5] Is DPO Superior to PPO for LLM Alignment? A Comprehensive Study. arXiv 2024
>
> [6] Unpacking DPO and PPO: Disentangling Best Practices for Learning from Preference Feedback. NeurIPS 2024
>
> [7] Weak-to-Strong Generalization: Eliciting Strong Capabilities With Weak Supervision. ICML 2024
>
> [8] The Unreasonable Effectiveness of Easy Training Data for Hard Tasks. ACL 202

---

> > ### Comment · Reviewer_Fiwv · 2024-11-22
> >
> > I appreciate your comprehensive response and the additional experimental results with KTO that support your claims. But I still find that the core concerns have not been fully addressed. The fundamental question about whether this is truly an RLHF pipeline issue or rather a human evaluation quality problem remains unclear. Your response seems to sidestep this distinction by focusing on the empirical evidence of the phenomenon itself, rather than its root cause.
> >
> > Thanks for the response, I stand by my original scores.

---

> > > ### Author Response · Authors · 2024-11-22
> > > **Response to Reviewer Fiwv**
> > >
> > > Thanks for your follow-up question!
> > >
> > > > The fundamental question about whether this is truly an RLHF pipeline issue or rather a human evaluation quality problem remains unclear.
> > >
> > > As we highlighted at the beginning of the abstract and introduction, limited human evaluation quality introduces this problem, and RLHF **exacerbates** this problem due to reward hacking. As indicated by our KTO experiments, other methods that optimize against human evaluation might also exacerbate this problem.

---

> > > > ### Author Response · Authors · 2024-11-24
> > > > **Looking forward to your comment**
> > > >
> > > > We have added further clarifications and experiments to address your reviews. Please feel free to let us know if you have any questions. Thanks a lot!!

---

### Author Response · Authors · 2024-11-18
**General Response**

Dear Reviewers,

Thank you all for your thoughtful reviews! We are excited that the reviewers recognized the strength of our paper:

- **Well-motivated, significant topic, novel framing, impactful insights (all reviewers)**:  “Addresses a critical gap in understanding how language models might naturally learn to mislead humans”
- **Well-designed, comprehensive experiments (all reviewers)**: “incorporate diverse and challenging tasks like question-answering and programming tasks,” and “The human evaluations are well thought out.”
- **Well-written, easy-to-follow (Fiwv, cFxf, 24F3)**:  “The paper is well-written and easy-to-follow.”

Please see our clarification and added experiments in individual responses.

---

### Meta-Review · Area_Chair_3TXj · 2024-12-16

**Metareview:**

The paper studies the phenomenon of “U-SOPHISTRY” where LLMs trained with RLHF unintentionally become more convincing to humans about incorrect outputs without improving actual performance. The work performs extensive experiments on QA and programming tasks, and shows that RLHF-optimized models persuade human evaluators of incorrect answers more often than pre-trained models.

This work is well motivated and provides systematic study of many pheromones mentioned by existing works such as impact of noisy annotations. The empirical validation, the design of human annotation is insightful and beneficial to the alignment community.

**Additional Comments On Reviewer Discussion:**

The authors provided further explanations and quantitative analysis on some unclear parts of the paper.

---

### Decision · Program_Chairs · 2025-01-22

Accept (Poster)